# Magnetic Nanoparticle-Based Dispersive Solid-Phase Microextraction of Three UV Blockers Prior to Their Determination by HPLC-DAD

**DOI:** 10.3390/ijerph19106037

**Published:** 2022-05-16

**Authors:** Suad E. Abughrin, Usama Alshana, Sezgin Bakirdere

**Affiliations:** 1Department of Analytical Chemistry, Faculty of Pharmacy, Near East University, TRNC, Mersin 10, Nicosia 99138, Turkey; ualshana@neu.edu.tr; 2Center for Solar Energy Research and Studies, Department of Renewable Energy, Libyan Authority for Scientific Research, Tripoli P.O. Box 30454, Libya; 3Department of Chemistry, College of Science, Sultan Qaboos University, 123 Al Khod, Muscat P.O. Box 50, Oman; 4Department of Chemistry, Faculty of Arts and Science, Yıldız Technical University, Istanbul 34349, Turkey; sezgin@yildiz.edu.tr

**Keywords:** dispersive solid-phase microextraction, HPLC-DAD, magnetic nanoparticles, sunscreen products, UV blockers

## Abstract

The need for proper handling of environmental samples is significant, owing to their environmental effects on both humans and animals, as well as their immediate surroundings. In the current study, magnetic nanoparticle-based dispersive solid-phase microextraction was combined with high-performance liquid chromatography using a diode array as the detector (HPLC-DAD) for both the separation and determination of three different UV blockers, namely octocrylene, ethylhexyl methoxycinnamate, and avobenzone. The optimum conditions for the extraction were found to be as follows: Stearic acid magnetic nanoparticles (20 mg) as the sorbent, acetonitrile (100 µL) as the eluent, as well as a sample pH of 2.50, adsorption and desorption time of 1.0 min, with a 3.0 mL sample volume. The limits of detection were as low as 0.05 µg mL^−1^. The coefficient of determination (R^2^) was above 0.9950, while the percentages of relative recoveries (%RR) were between 81.2 and 112% for the three UV blockers from the environmental water samples and sunscreen products.

## 1. Introduction

The increase in industrial developments globally has resulted in a rapid decrease in the stratospheric ozone by a magnitude of 1% yearly, leading to a reduction in the capacity of the ozone layer to absorb and isolate solar ultraviolet (UV) radiation [1]. Long-term exposure to this radiation causes substantial harmful effects on the human skin. These concerns have raised awareness of the need for sun protection, leading to the popularity of sunscreen products, which are widely recommended by dermatologists for preventing skin damage caused by UV radiation [2]. However, the active ingredients in these sunscreen products, popularly known as UV blockers, have been shown to have several dermatological complications such as dermatitis and allergies, prompting regulatory restrictions to be placed on the maximum permissible concentration that can be used in sunscreen products [3]. The major regulatory agencies, such as the United States (US) Food and Drug Administration (FDA), the European Union (EU) Cosmetics Directive (CD), and Japanese legislation, allow a maximum of 10% of organic UV blockers to be used in cosmetics [4]. Even though these analytes are present in high concentrations in sunscreen products, an efficient sample cleanup step is critical to reduce the matrix effect and preserve column life due to the highly oily nature of such samples. Moreover, there are no official analytical methods dedicated to the determination of UV blockers in sunscreen products [5]. As a result, sensitive and efficient analytical procedures should be developed to ensure that these limits are met to guarantee human safety. Consequently, several methods have been proposed in the literature for the determination of UV blockers in cosmetics and a detailed review has been published earlier [5]. Nevertheless, the most common technique that has been used for this purpose is liquid chromatography (LC) with ultraviolet/visible (UV/Vis) spectrometry detection due to the high absorbance capacity of these compounds in the UV region of the spectrum [6].

Recently, the concern about the use of UV blockers has extended from the risk associated with direct exposure of these chemicals to humans through the use of sunscreen products to environmental concerns due to the accumulation of UV blockers in the aquatic environment through direct sources such as swimming, bathing, and industrial discharge and indirect sources such as wastewater discharge from wastewater-treatment plants [7]. This can lead to the formation of by-products that can potentially be more toxic than the original compounds when these UV blockers react with other compounds in the environment and sunlight [8]. In line with these concerns, it is imperative to monitor the concentration of these compounds in the aquatic environment to halt their bioaccumulation in the environment by developing fast and sensitive methods that can be used to determine UV blockers present in environmental materials. The determination of UV blockers in environmental and water bodies often relies on a microextraction step in order to preconcentrate the analytes in the sample since they are present at low concentrations, and a detailed review has been written previously on the use of both solvent-based and sorbent-based microextraction techniques for the determination of UV blockers in environmental water samples [9].

Sorbent-based microextraction, first introduced in 1990 and termed solid-phase microextraction (SPME), was derived from conventional solid-phase extraction (SPE) as a solvent-free sample preparation technique that integrates sampling, isolation, and concentration [10]. In SPME, the analytes were extracted from a sample solution by adsorption on the surface of a fused-silica fiber core that was coated with a suitable material selective to the analyte [11]. The analyte can be desorbed directly into a gas chromatograph (GC) [12] or liquid chromatography [13]. Dispersive solid-phase microextraction (DSPME) was later developed as a remedy for the limitations of SPME associated with the fiber core that was used, such as the duration needed for coating the fiber with a suitable sorbent, the swelling of the sorbent when exposed to certain organic solvents, the instability of the sorbent on the surface of the core, and the fragility of the fiber [14]. The mechanism of DSPME depends on dispersing the adsorbent in the aqueous sample to increase the surface area of contact between the sorbent and sample solution to enhance the extraction efficiency of the analyte, which is subsequently desorbed by a suitable solvent [15]. Recently, the use of magnetic nanoparticles (MNP) for DSPME, called magnetic dispersive solid-phase microextraction (MDSPME), has been proposed to facilitate the collection of the sorbent from the sample solution, which is a major challenge of conventional DSPME, by using an external magnetic field [16,17]. This enables the elimination of the centrifugation step [14]. Another development in MDSPME aimed at improving the selectivity of the method is the modification of the surface of the sorbent with a molecule such as citric acid [18], stearic acid (SA) [19], or molecularly imprinted polymers (MIPs) [20].

Several variations of DSPME have been proposed for the determination of UV blockers in environmental water samples based on magnetic graphitized carbon black [21], oleic acid-coated MNPs [22], etc. The hyphenation of MDSPME with another microextraction technique also occurs, such as stir bar sorptive extraction (SBSE), termed stir bar sorptive-dispersive microextraction (SBSDME) [23,24], as well as cloud-point extraction (CPE), termed cloud point–dispersive µ-solid phase extraction (CP-D-µ-SPE) [25]. MDSPME extraction has also been used as a tool for the retrieval of the extraction solvent for homogeneous liquid–liquid microextraction (HLLME), termed magnetic retrieval of the switchable hydrophilicity solvent (MR-SHS-HLLME) [26], and ionic liquid dispersive liquid–liquid microextraction (IL-DLLME) [27]. Although the performance of MDSPME is improved with these modifications, the combination with other microextraction techniques increases the sample preparation steps that can increase the probability of error as well as the overall analysis time.

In this regard, stearic-acid-modified magnetic solid-phase microextraction (SA-MDSPME) was used for the first time based on our current knowledge for the extraction and sample cleanup of three UV blockers, i.e., 2-ethylhexyl 2-cyano-3,3-diphenylacrylate (octocrylene, OCT), butyl methoxy dibenzoyl methane (avobenzone, AVO), and ethyl hexyl methoxycinnamate (EMC), from both sunscreen products and environmental water samples (swimming pool, tap, and sea) prior to separation and detection by HPLC-DAD. Influential experimental parameters affecting the performance of the SA-MDSPME- HPLC-DAD method, including the type and amount of the sorbent, sample pH, the type and amount of eluent, sample volume, as well as the adsorption and desorption time were optimized. This study presents the use of a cost-effective, easy, eco-friendly, and fast method due to the use of a modified MNP as a sorbent, which enabled the elimination of the centrifugation step.

## 2. Materials and Methods

### 2.1. Chemicals and Reagents

All the reagents and chemicals used in the current work were of analytical grade. AVO (log*P* 4.56, p*K*_a_ 9.74), OCT (log*P* 6.78), EMC (log*P* 5.38), acetic acid, disodium phosphate, HPLC-grade acetonitrile (ACN), ethanol (EtOH), methanol (MeOH), tetrahydrofuran (THF), monosodium phosphate, and sodium acetate were procured from Sigma-Aldrich (Darmstadt, Germany). Trifluoroacetic acid (TFA) was obtained from Fluka (Dresden, Germany). Stearic acid-modified MNPs synthesis and characterization were carried out previously by the group [28]. The preparation of aqueous solutions was performed using deionized (DI) water (18.2 MΩ-cm), obtained with Pure Lab Ultra Analytic (ELGA Lab Water, Lane End, UK). For the calculation of p*K*_a_ and log*P* values (i.e., the logarithm of octanol/water partition coefficient), Marvin Sketch (Rev. 20.11.0, ChemAxon Ltd., Boston, MA, USA) was used.

### 2.2. Synthesis of Coated MNPs

The synthesis of stearic acid-coated nanoparticles was carried out according to the method published in the literature, where nanoparticles were coated with different coating materials (Wang et al. 2011; Clare 2016). Briefly, 2.703 g of FeCl_3.6_H_2_O and 1.961 g of (NH_4_)_2_Fe(SO_4_)_2.6_H_2_O were dissolved in 100.0 mL of ultrapure deionized water in a flask, heated to 80 °C, and maintained there with constant and vigorous stirring. One milliliter of a concentrated stearic acid solution prepared in ethanol and 10 mL of 25% ammonia solution were added dropwise to the mixture, resulting in a color change from a brick color to a blackish gray color. The reaction was carried out under nitrogen gas to provide an inert atmosphere for 2.0 h at 80 °C. Stearic-acid-coated MNPs were separated with a neodymium magnet and washed with ethanol and ultrapure water several times to eliminate excess acid and then dried at 50 °C for 24 h [28].

### 2.3. Standard Solutions of UV Blockers

Three separate stock solutions containing 1000 µg mL^−1^ of AVO, OCT, and EMC were prepared in ACN and stored at −15 °C until required. The mixture of the solutions of these three compounds was prepared by diluting the original stock solution to the desired concentration, with ACN as a working standard solution. The working standard solution was re-diluted for calibration purposes.

### 2.4. Apparatus and Conditions

Chromatographic separation was carried out using an Agilent Technologies 1200 Series (USA) HPLC chromatograph, monitored with an Agilent ChemStation for LC 3D systems (Rev. B.03.01) software equipped with an autosampler, degasser, quaternary pump, thermal column jacket, and DAD, with a reversed-phase column (Zorbax 4.6 mm ID × 150 mm, 5 µm, Agilent Technologies, Waldbronn, Germany), using an isocratic elution consisting of 80/20 (%*v*/*v*) MeOH:0.5% TFA in DI water with a 0.9 mL min^−1^ flow rate. A 40 °C column jacket temperature was set with a 20 µL injection volume. The DAD was set at the maximum absorption wavelength of the analytes (310 nm for OCT and EMC, 360 nm for AVO).

Water samples were filtered using vacuum filtration through a 0.45 µm cellulose membrane (Whatman, Dassel, Germany) and 0.2 µm sterile nylon syringe filters (Chromfil, China). A FiveEasy Plus^TM^ pH meter was used for all pH measurements. Vortex mixing was performed by an MS 3 digital vortex (IKA, Staufen, Germany). Centrifugation was carried out with EBA20 Portable Centrifuge C2002 (Hettich, Kirchlengern, Germany). Micropipettes (10–100 µL and 100–1000 µL) were obtained from Sigma-Aldrich (St. Louis, MI, USA). For the weighing of the samples, an electronic balance (Mettler-Toledo, Greifensee, Switzerland) was used.

### 2.5. Sample Preparation

The environmental sample (Swimming pool, tap, and seawater samples) were collected from three different locations in the Turkish Republic of Northern Cyprus (TRNC) in 1.0 L plastic bottles and were stored at room temperature. Double filtration was carried out with 0.45 µm Whatman filter paper and 0.2 µm sterile nylon syringe filters prior to analysis. Sunscreen products were obtained from local stores and pharmacies in Nicosia, TRNC. A 20 mg portion of the sunscreen samples was weighed and dissolved with 1.0 mL of ACN in a 15 mL screw-capped, graduated, polypropylene centrifuge tube by vortex for 4 min and centrifugation for 1 min at 6000 rpm. The solution was diluted 500 times with DI water and taken as the sample solution for the sunscreen. The overall schematic diagram of the proposed MNP-DSPME-HPLC-DAD procedure is shown in (Figure 1).

### 2.6. Magnetic Nano-Particles Dispersive-Solid Phase Microextraction

A 2.0 mL portion of the sample solution (environmental and sunscreen) was added to a 15 mL screw-capped, graduated, polypropylene centrifuge tube and buffered with 1.0 mL of phosphate buffer solution (pH 2.5). The solutions were transferred into 20.0 mg of preconditioned SA-MNPs as the adsorbent and vortexed for 1 min. An external magnetic field was used to collect the analyte-rich SA-MNPs, and the supernatant was discarded. Regarding the external magnetic field, Reyes-Gallardo et al. [29] reported that this property is essential in magnetic solid-phase extraction since it allows the easy dispersion of MNPs in aqueous or organic media and their simple recovery by applying a magnetic field, usually in the form of an external magnet. Moreover, according to Lu et al. [30], in biotechnology and biomedicine, magnetic separation can be used as a quick and simple method for the efficient and reliable capture of specific proteins or other biomolecules. Most particles currently used are super paramagnetic, meaning that they can be magnetized with an external magnetic field and immediately dispersed once the magnet is removed. The analyte desorption was performed using 100 μL of ACN as the eluent by vortex for 1 min. The collected solution was diluted two times with DI water, and 20 µL was injected into HPLC.

## 3. Results and Discussion

### 3.1. Optimization of SA-MDSPME Conditions

The parameters that are critical to achieving maximum extraction efficiency in SA-MDSPME were optimized, including the type and amount of sorbent, in which three types were used, including SA-MNP, Zirconium nanoparticles (Zr-NP), and multi-walled carbon nanotubes (MW-CNT) ranging from 10–50 mg. Furthermore, the pH of the sample solution was investigated using different buffer solutions ranging from 2.5–6.5, the type and amount of eluent were evaluated using different organic solvents (ACN, Et-OH, MeOH, and THF) with different volumes within the range of 50–400 mL, the adsorption and desorption time were investigated within 0–4.0 min, and the sample volume was investigated by varying the volume from 2–6 mL. The influence of each parameter on the method’s extraction efficiency was studied by changing a single parameter at a time while keeping the remaining parameters constant. The absolute percentage recovery (%R) was used to determine the efficiency of the extraction.

#### 3.1.1. Type and Amount of Sorbent

A critical parameter for the extraction and preconcentration of the target analyte is the selection of the appropriate sorbent. The impact of the type of sorbent on the recovery was evaluated using three different sorbents, SA-MNP, Zr-NP, and MW-CNT. The highest recovery was observed using SA-MNP (Figure 2A), which could be due to the combined effect of the modification of the magnetic nanoparticles with the stearic acid functional group that can increase the preconcentration and selectivity of the adsorbent to the analytes [28] and the magnetic property of the sorbent that permits easy separation of the sorbent from the sample solution using an external magnetic field in contrast to their non-magnetic alternatives used in the study, thereby eliminating the centrifugation step in addition to preventing any blockage from occurring in the HPLC capillary tubes and column.

Generally, the analyte preconcentration is influenced by the amount of sorbent material used. To investigate the effect of the amount of SA-MNPs on the recovery of OCT, AVO, and EMC, various amounts of sorbent ranging from 10–50 mg were applied. The average recovery increased linearly with an increase in the amount of sorbent from 10 to 20 mg due to an increase in the sorbent’s surface area. Therefore, it resulted in more interaction between the analytes and SA-MNP. However, the recovery decreased and remained stable in the range of 30 to 50 mg due to the insufficient volume of ACN (100 µL) to elute the retained analytes in sorbent (Figure 2B). Therefore, 20 mg of SA-MNP was used as the optimum amount of sorbent.

#### 3.1.2. Sample pH

Sample pH is considered significant in terms of extraction efficiency and recovery owing to its impact on the affinity of coated nanoparticles with the analyte [31]. pH levels in the range of 2.5–6.5 were evaluated for the recovery of the analytes. This was achieved by using different buffer solutions (phosphate and acetate). At a higher pH, the analytes can be hydrolyzed, leading to a decrease in the extraction efficiency and, hence, low recovery [22,24]. The highest recovery was obtained at pH 2.5 (Figure 2C) and was used for all analyses.

#### 3.1.3. Selection of the Type and Volume of Eluent

A suitable eluent that can desorb the analyte from the surface of the sorbent is desired. For this reason, ACN, EtOH, MeOH, and THF were selected for this study. For the three UV blockers examined, ACN gave the highest recovery (Figure 2D) and was used for the rest of the experiments. The volume of ACN as the eluent was investigated within the range of 50–400 µL. The results demonstrated that the highest %R of the analytes were observed with 100 µL of ACN before decreasing, which could be a result of the dilution of the analyte due to the excess volume of ACN (Figure 2E). However, the minimal volume of eluent (below 100 μL) would not be enough for elution of all the analytes and may lead to low recovery of the analytes. Therefore, 100 μL of ACN was taken as the optimum volume of the eluent.

#### 3.1.4. Adsorption and Desorption Time

To achieve a high degree of interaction between the sample solution and the sorbent, the adsorption time was considered as the time period between the addition of the sample solution to the SA-MNP sorbent and the moment just before detection, which corresponds to the time of vortexing. The effect of the duration of the adsorption was investigated with a vortex time of 0–4.0 min. It was observed that applying 10 s of manual shaking prior to vortexing was insufficient for the distribution of particles throughout the sample, hence a minimum vortex time of 1.0 min will accelerate the contact between the analytes and the sorbent, leading to an increase in recovery. A further increase in the contact time did not affect recovery, signaling that equilibrium was achieved at 1.0 min (Figure 2F). Therefore, the optimum adsorption time was selected as 1.0 min. To study the impact of the desorption time on analyte recoveries, the vortex time was applied once again between 0 and 4 min as shown in Figure 2. There was an increase in the recovery from 0 min (without vortex) to 1.0 min; the recovery increased due to agitation of the analytes by vortex to transfer them from the sorbent into the eluent. However, the trend remained unchanged from 1.0 min to 4.0 min. Hence, 1.0 min was considered the optimum desorption time to elute the analytes.

#### 3.1.5. Sample Volume

The influence of the sample volume on the recovery of analytes was investigated by varying the volume from 2.0–6.0 mL, while the concentration of the analyte was maintained at 1.0 µg mL^−1^ (see Figure 2H). This shows that the recoveries of three analytes increased up to 3.0 mL, after which they declined, likely due to the maximum sorbent capacity of 20 mg with the sample volume of 3.0 mL. Subsequently, 3.0 mL was employed as the maximum sample volume.

### 3.2. Analytical Performance

An aqueous calibration graph was used to assess the performance of the proposed method by preparing a standard solution of UV blockers between the range of 0.0 to 15.0 µg mL^−1^ without microextraction prior to separation and detection by HPLC-DAD. Furthermore, the environmental samples were spiked with a standard solution of UV-blockers between 0.0 and 6.5 µg mL^−1^, while the sunscreen products were spiked between 0.0–5.5 µg mL^−1^ before applying the optimized MNPs-DSPME-HPLC-DAD procedure to plot the standard addition calibration graphs. From the results shown in Table 1, calibration curves with a determination coefficient (R^2^) between 0.9952 and 0.9996 were observed, as well as detection limits (LOD) within 0.31 to 0.43 µg mL^−1^ for the aqueous standard, 0.07 to 0.18 µg mL^−1^ for the environmental samples, and 0.05 to 0.14 µg mL^−1^ for the sunscreen products, which were determined by applying equation 3 (S_b_/m), whereby m is considered the slope while S_b_ is the standard deviation of the regression equation. The limit of quantitation (LOQ), calculated based on equation 10 S_b_/m, was within the range of 1.03 to 1.43 µg mL^−1^ for the aqueous standard, 0.22 to 0.61 µg mL^−1^ for the environmental samples, and 0.16 to 0.48 µg mL^−1^ for the sunscreen products. A linear response was attained from LOQ to 15 µg mL^−1^ for the aqueous calibration graph, LOQ to 6.5 µg mL^−1^ for the environmental samples, and LOQ to 5.5 µg mL^−1^ for the sunscreen products. The method’s precision was assessed using percentage relative standard deviation (%RSD) with intraday precision between 3.05 and 3.72% for the aqueous standard, 2.34 and 6.90% for environmental samples, and 1.12 and 3.68% for the sunscreen products. The inter-day precision was between 5.05 and 6.65% for the aqueous standard, 3.83 and 13.21% for environmental samples, and 2.53 and 8.74% for the sunscreen products.

### 3.3. Effect of Matrix and Addition-Recovery Studies

To be able to evaluate the possible matrix effect, an addition–recovery study was carried out by spiking environmental and sunscreen samples with standards of UV-blockers at three levels of concentration (0.5, 3.5, and 5.0 µg mL^−1^ for environmental samples and 1.5, 3.5, and 4.5 µg mL^−1^ for sunscreen samples) and applying the proposed SA-MDSPME-HPLC-DAD method. The percentage of relative recovery (%RR) was found to be between 81.2 and 112 (see Table 2). After a comparison between different slopes of the calibration curves, statistical analysis using ANOVA was applied to evaluate the effects of the matrix (see Table 2). A statistical difference was observed (*p* < 0.05), which was an indication of the effect of the matrix, making it necessary to apply the standard-addition method to be able to eradicate it. The sunscreen samples were found to contain the UV blockers stated in the label, with sunscreen 1 containing EMC at a concentration of 3.3%, *w*/*w*, sunscreen 2 containing AVO at a concentration of 6.4%, *w*/*w,* and finally, sunscreen 3 containing OCT and AVO at a concentration of 4.6 and 4.3%, *w*/*w*, respectively. The analytes were undetected in the environmental samples, likely due to the samples being collected during the COVID-19 pandemic when outdoor activities were limited.

### 3.4. Comparison with Other Studies

The proposed SA-MDSPME-HPLC-DAD method was compared with similar studies in the literature for the determination of UV blockers in sunscreen products and environmental samples in terms of analysis time, the volume of organic solvent, detection limit, and precision based on the percentage of relative standard deviation.

The proposed study showed superior performance compared to those in the literature for the determination of the analytes in sunscreen products by requiring the least analysis time and the least volume of organic solvent, except for [6], which required only 2 min.

On the other hand, the sensitivity (LOD) was similar to other methods found in the literature (see Table 3). Comparing the current proposed study for the determination of environmental water samples with other recent studies conducted in the literature, a lower consumption of the organic solvent, as well as a shorter analysis time, was observed. In terms of sensitivity, the techniques in the literature exhibited better sensitivity, although the analytes could not also be detected or quantified in the environmental samples for those techniques except for in [32]. This is because the analytes are present in very low concentrations in environmental samples; therefore, strong sensitive analytical instruments are required.

## 4. Conclusions

In the current work, SA-MDSPME was used for the extraction of UV blockers and sample cleanup of environmental samples and sunscreen products prior to separation and detection by HPLC-DAD. The method decreased the use of organic solvents substantially in addition to fast sample preparation due to the use of magnetic nanoparticles. This can simplify the microextraction step and decrease the energy consumption by replacing the centrifugation step with a magnetic field for the separation of the sorbent from the sample solution and eluting the analyte, which agrees with green chemistry protocols. In addition, the microextraction served as a sample cleanup, which is especially important when oily samples are analyzed to protect and increase the lifetime of the column. The analyte could be detected in genuine sunscreen samples and could serve as a useful alternative for these kinds of analyses. The findings of the current study showed that the optimum conditions for the extraction step were found to be as follows: SA-MNPs (20 mg) as the sorbent, ACN (100 µL) as the eluent, along with a sample pH of 2.50, adsorption and desorption times of 1.0 min, with a 3.0 mL sample volume. The LOD was as low as 0.05 µg mL^−1^. The coefficient of determination (R^2^) was above 0.9950, while the percentage of relative recoveries (%RR) was within 81.2–112% for the three UV blockers from the environmental water samples and sunscreen products.

## Figures and Tables

**Figure 1 ijerph-19-06037-f001:**
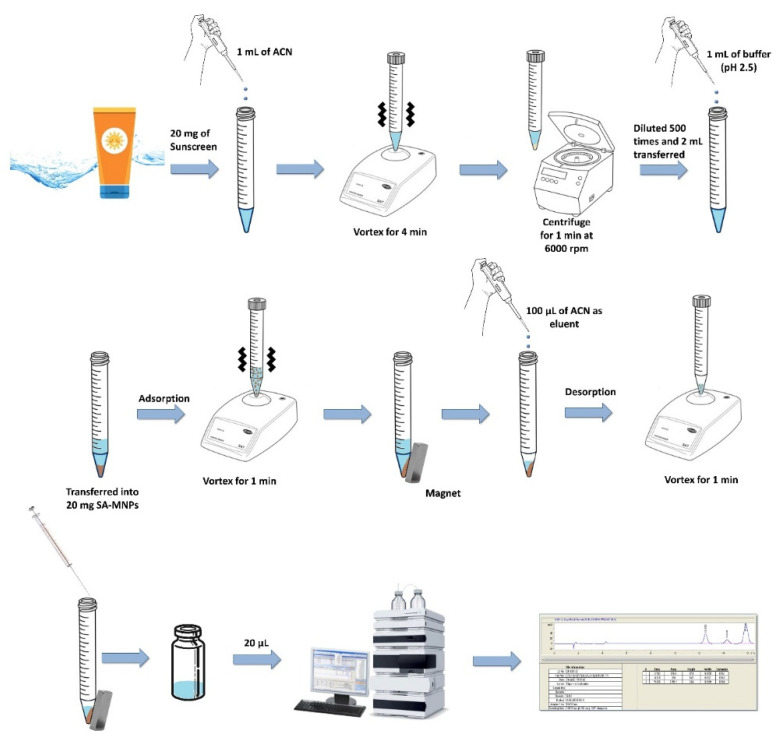
Schematic diagram of the proposed MNP-DSPME-HPLC-DAD procedure.

**Figure 2 ijerph-19-06037-f002:**
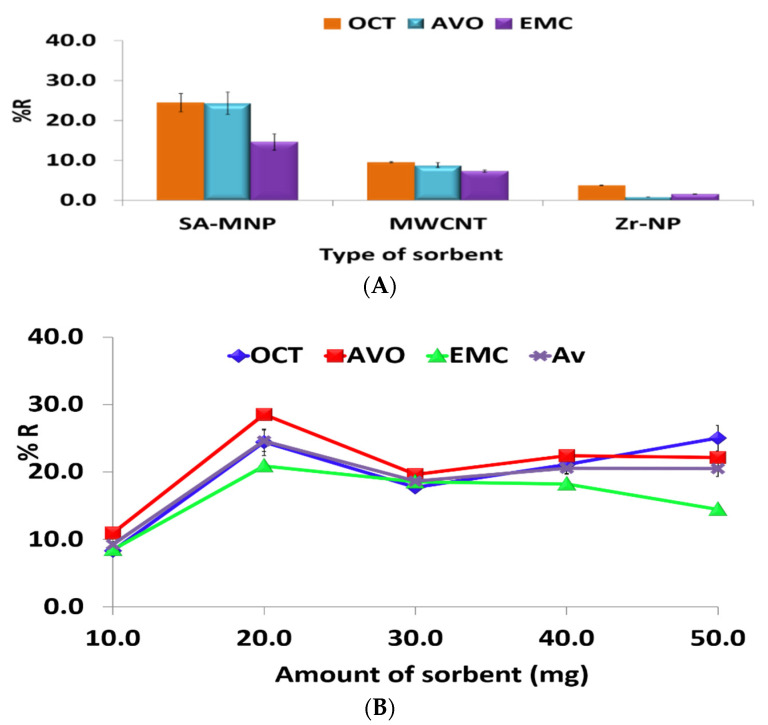
Optimization of the DSPME procedure: (**A**) Type of sorbent; (**B**) amount of sorbent; (**C**) sample pH; (**D**) type of eluent; (**E**) volume of eluent; (**F**) adsorption time; (**G**) desorption time; and (**H**) sample volume. Chromatographic conditions: RP-HPLC (Zorbax 4.6 mm ID × 150 mm (5 µm), isocratic elution consisting of 80/20 (%*v*/*v*) MeOH:0.5% TFA in DI water, 0.9 mL min^−1^ flow rate, 40 °C column temperature, and injection volume of 20 µL.

**Table 1 ijerph-19-06037-t001:** Analytical performance of DSPME-MNPs-HPLC-DAD for environmental and sunscreen samples.

Method	Sample	Analyte ^a^	Regression Equation ^b^	R^2^	RSD ^c^		LOD ^d^	LOQ ^e^	LDR ^f^
					Intraday	Interday			
HPLC-DAD	Aq.	OCT	y=44.9(±0.5)x−12.9(±4.6)	0.9975	3.05	5.05	0.31	1.03	1.03–15
		AVO	y=89.6(±1.4)x−26.3(±12.9)	0.9952	3.18	5.16	0.43	1.43	1.43–15
		EMC	y=107.7(±1.6)x−39.8(±14.2)	0.9959	3.72	6.65	0.40	1.32	1.32–15
	Pool 1	OCT	y=247.9(±2.2)x+32.7(±8.2)	0.9987	3.56	7.27	0.10	0.33	0.33–6.5
		AVO	y=613.1(±3.7)x+20.2(±13.8)	0.9994	2.34	3.83	0.07	0.22	0.22–6.5
		EMC	y=556.4(±6.9)x−50.2(±25.6)	0.9975	4.60	8.45	0.14	0.46	0.46–6.5
	Pool 2	OCT	y=254.4(±4.1)x+12.4(±15.5)	0.9957	5.32	7.40	0.18	0.61	0.61–6.5
		AVO	y=566.2(±6.9)x+45.4(±25.9)	0.9976	6.84	14.43	0.14	0.46	0.46–6.5
		EMC	y=551.7(±6.6)x−61.0(±24.9)	0.9976	3.80	7.13	0.14	0.45	0.45–6.5
	Pool 3	OCT	y=347.5(±5.2)x−22.1(±19.4)	0.9964	3.44	5.13	0.17	0.56	0.56–6.5
		AVO	y=802.4(±11.8)x−72.7(±44.0)	0.9965	2.46	4.05	0.16	0.55	0.55–6.5
		EMC	y=752.8(±9.2)x−74.3(±34.3)	0.9976	4.22	7.63	0.14	0.46	0.46–6.5
	Tap	OCT	y=301.2(±4.1)x+22.7(±15.4)	0.9970	6.90	13.21	0.15	0.51	0.51–6.5
		AVO	y=720.6(±7.8)x+32.0(±29.2)	0.9981	5.90	12.82	0.12	0.40	0.40–6.5
		EMC	y=637.1(±7.1)x+39.1(±26.5)	0.9980	5.68	12.48	0.12	0.42	0.42–6.5
	Sea	OCT	y=342.8(±3.7)x−25.9(±13.6)	0.9981	3.12	3.12	0.12	0.40	0.40–6.5
		AVO	y=775.8(±12.3)x−24.1(±46.1)	0.9959	5.66	5.66	0.18	0.59	0.59–6.5
		EMC	y=682.9(±5.0)x−64.1(±18.8)	0.9991	2.94	2.94	0.08	0.28	0.28–6.5
	Sunscreen 1	EMC	y=400.6(±4.5)x+526.8(±15.4)	0.9980	2.68	4.68	0.12	0.38	0.38–5.5
	Sunscreen 2	AVO	y=207.6(±0.9)x+528.2(±3.2)	0.9996	1.12	2.53	0.05	0.16	0.16–5.5
	Sunscreen 3	OCT	y=167.1(±1.8)x+305.5(±6.4)	0.9980	2.26	3.67	0.12	0.38	0.38–5.5
		AVO	y=329.0(±4.6)x+602.1(±15.8)	0.9968	3.68	8.74	0.14	0.48	0.48–5.5

^a^ OCT: Octocrylene, AVO: Avobenzone, EMC: 2-Ethylhexyl-4-methoxycinnamate; ^b^ Peak area = slope(±SD) × [concentration (μg mL^−1^) + intercept(±SD). ^c^ Percentage relative standard deviation, *n* = 3. ^d^ Limit of detection (μg mL^−1^); ^e^ Limit of quantitation (μg mL^−1^); ^f^ Linear dynamic range (μg mL^−1^).

**Table 2 ijerph-19-06037-t002:** Percentage of relative recoveries of UV blockers from environmental and sunscreen samples.

Sample	Added (μg mL^−1^)	Found (μg mL^−1^)			%RR ^a^		
OCT	AVO	EMC	OCT	AVO	EMC
Pool 1	-	<LOD	<LOD	<LOD	-	-	-
	0.5	0.5	0.5	0.6	108.8	103.9	111.0
	3.5	3.5	3.4	3.3	101.3	98.5	94.8
	5	4.9	4.9	4.9	98.7	98.6	98.7
Pool 2	-	<LOD	<LOD	<LOD	-	-	-
	0.5	0.4	0.4	0.4	92.2	80.4	91.6
	3.5	3.3	3.6	3.2	95.7	105.0	93.9
	5	5.3	5	5	105.2	99.1	99.6
Pool 3	-	<LOD	<LOD	<LOD	-	-	-
	0.5	0.4	0.5	0.4	96.7	99.9	90.2
	3.5	3.3	3.4	3.4	95.0	97.0	95.8
	5	5.3	4.8	5.2	105.3	95.7	104.0
Tap	-	<LOD	<LOD	<LOD	-	-	-
	0.5	0.4	0.4	0.4	82.0	81.8	82.7
	3.5	3.7	3.6	3.7	105.8	103.4	105.2
	5	5.1	5.1	5	101.5	101.8	100
Sea	-	<LOD	<LOD	<LOD	-	-	-
	0.5	0.4	0.5	0.4	95.9	106.0	90.2
	3.5	3.5	3.7	3.4	99.7	106.0	96.9
	5	5.1	5.1	4.9	103.4	101.8	99.7
Sunscreen 1	0	-	-	1.3 (3.3% *w*/*w*)	-	-	-
	1.5	-	-	1.4	-	-	96.7
	3.5	-	-	3.4	-	-	96.0
	4.5	-	-	4.5	-	-	99.8
Sunscreen 2	0	-	2.5 (6.4% *w*/*w*)	-	-	-	-
	1.5	-	1.6	-	-	103.5	-
	3.5	-	3.5	-	-	99.6	-
	4.5	-	4.5	-	-	99.7	-
Sunscreen 3	0	1.8 (4.6% *w*/*w*)	1.8 (4.6% *w*/*w*)	-	-	-	-
	1.5	1.5	1.7	-	101.8	112.6	-
	3.5	3.6	3.5	-	102.4	101.0	-
	4.5	4.4	4.4	-	99.0	97.5	-

^a^ Percentage relative recovery: A value calculated according to extraction yields obtained from standard-addition calibrations.

**Table 3 ijerph-19-06037-t003:** Comparison of DSPME-MNPs-HPLC-DAD with previous techniques in order to determine UV blockers in environmental and sunscreen samples.

Sample Type	Method ^a^	Analysis Time (min)	V_org_ ^b^ (mL)	LOD ^c^ (μg mL^−1^)	%RSD ^d^	Ref.
Water	TF-SPME-HPLC-UV	2.8	0.3	0.001–0.008	3–23	[33]
	CPE-D-µ-SPE-LC-DAD	9	0.250	0.0014–0.0075	4.5–14.9	[25]
	DLLME-HPLC-UV	14	0.015	0.0019–0.0064	1.9–8.0	[32]
	LOV-BI-LC	9	1.55	0.00045–0.0032	12.0–13.0	[34]
	DµSPE-LC-UV/Vis	25	5	0.0024–0.031	1.0–11.0	[24]
	DSPME-HPLC-DAD	2	0.1	0.07–0.18	6.1–9.5	This study
Sunscreen	HPLC-UV/Vis	30	100	0.01–1.99	0.16–12.69	[35]
	HPLC-UV/Vis	10	95–195	0.50–1.50	0.97–6.1	[36]
	HPLC-DAD	40	14.8	0.3	0.6–3.7	[37]
	LC-UV/Vis	<2	10	0.02–0.22	0.2–8.2	[6]
	DSPME-HPLC-DAD	6	1.1	0.05–0.14	6.1–9.5	This study

^a^ TF-SPME-HPLC-UV: Thin-film solid-phase microextraction–high-performance liquid chromatography–ultraviolet/visible detection. CPE-D-µ-SPE-LC-DAD: Cloud point-dispersive micro-solid phase extraction–liquid chromatography–diode array detection. DLLME-HPLC-UV: Dispersive liquid–liquid microextraction–high-performance liquid chromatography–ultraviolet/visible detection. LOV-BI-LC: Lab on valve-bead injection–liquid chromatography. DµSPE-LC-UV/Vis: Dispersive micro solid-phase extraction–liquid chromatograph–ultraviolet/visible detection. HPLC-UV/Vis: High-performance liquid chromatography–ultraviolet/visible detection; HPLC-DAD: High-performance liquid chromatography–diode array detection; LC-UV/Vis: Reversed-phase–liquid chromatography–ultraviolet/visible detection; ^b^ Total volume of organic solvent consumed per sample. ^c^ Limit of detection. ^d^ Percentage relative standard deviation.

## Data Availability

The data used in this study will be provided on request.

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
