# Peer review of "Magnetic Nanoparticle-Based Dispersive Solid-Phase Microextraction of Three UV Blockers Prior to Their Determination by HPLC-DAD"

_ijerph, 2022, doi:10.3390/ijerph19106037_

Round 1

Reviewer 1 Report

The authors discussed a topic in the analytical techniques which is  determination of three UV blockers (octocrylene, ethylhexyl methoxycinnamate, and avobenzone) from swim pools water and sun screen cream using magnetic nanoparticle-based dispersive solid-phase microextraction  hyphenated with HPLC-DAD.

I would to ask authors:

In the  abstract part:

How is the percentage relative recoveries (%RR) were within 81.2- 18
112%? how is the percent exceed 100%

The authors should write a note about  SA-MNPs  source and preparation not add reference only.

line 189, the word min should not bold

Reviewer 2 Report

I have reviewed the manuscript and I suggest a revision before its publication. (1) The abstract needs to be rewritten in a clear and more detailed way.
(2) Figure 2 needs to be clearer with better resolution.
(3) In line 143 what does tap sea water mean? 
(4) There are reference errors in many places. 

Reviewer 3 Report

Subject of the study is practical and interesting, however some points are not adequately clarified and some aspects of the experiment should be broaden. Detailed remarks are presented below.

Why only one dilution of 500 times was chosen? Some additional dilutions should be added.

It is not clear how many sources of the water were analyzed finally. Did they were analyzed before sunscreen addition?

Procedure presented in Fig. 1 is not completely described in the text.

Parameters of the external magnetic field are not presented. How it was optimized?

How many samples were analyzed? What was the repeatability?

Standard and environmental samples procedures should be described separately and more adequately.

Ranges of all parameters investigated and presented in chapter 3.1 (type and amount of sorbent, pH of sample solution, type and amount of eluent, sample volume, adsorption and desorption time) should be written in text with justification for the selection.

How the uncertainty was determined in Fig. 2?  Are the changes statistically significant?

What were the limitations of the comparison with the other studies?

Conclusions should be more related to data presented in the abstract.

Manuscript requires editorial correction.

Round 2

Reviewer 3 Report

Although the manuscript has been improved I still have some minor comments, presented below.

If not a detailed description related to magnetic field, at least the proper citation should be added in the text.

It is still not clear how the uncertainty presented in Fig. 2 was calculated? And any statistical test was not applied to clarify if the difference are statistically significant.

Manuscript requires editorial correction for newly added parts (abstracts, figures).
